# Precision Without Labels: Detecting Cross-Applicants in Mortgage Data Using Unsupervised Learning

## Abstract

We propose a novel method for evaluating unsupervised anonymous record linkage without requiring labeled training data. We derive observable lower bounds on both precision and relative recall by exploiting a common structural constraint that limits how many positive outcomes a single individual can have. This enables principled tuning and comparison of label-generating models without labeled training data. We demonstrate the method on Home Mortgage Disclosure Act (HMDA) data, using a clustering algorithm to detect loan applicants who submit multiple applications ("cross-applicants") in a dataset lacking personal identifiers. Our preferred specification identifies cross-applicants with 92.3% precision with only minimal loss in relative recall.

## 1 Introduction

We derive observable lower bounds on both precision and relative recall in unsupervised ML-based anonymous record linkage settings. Our framework is domain-agnostic and requires only three features of the data: (1) records (rows) that correspond to single transactions (e.g., mortgage applications), (2) multiple records that may belong to the same individual, and (3) a structural constraint that limits how many positive outcomes a single individual can have (e.g., individuals can originate only one first-lien mortgage). Similar structural constraints are common, and examples include: i) secured loan applications (each person originates at most one loan); ii) insurance quotes (each person takes out at most one policy per category) ; iii) college applications (each person attends at most one college); or iv) job applications (each person accepts at most one full-time offer).

Such data often lack personal identifiers, especially when data is cross-institutional or privacy-constrained. In all cases, our results can be used to link the records from the same individual. Further, our theory is method-agnostic: Because our theoretical bounds depend only on predicted labels, they apply to any algorithm that generates such labels. Consequently, they enable both hyper-parameter tuning and cross-model comparisons. To our knowledge, this is the first work to derive observable lower bounds on both precision and relative recall in unsupervised classification settings, and to show how these can be used to tune or compare models in the absence of labels.

Our running example, and application, is the detection of "cross-applicants" – individuals who submit multiple mortgage applications – using a clustering-based algorithm applied to the Home Mortgage Disclosure Act (HMDA) data. HMDA is a leading example of a consumer finance dataset at the account level that does not include person-level identifiers: it contains the near-universe of all mortgage applications in the US, but does not contain an applicant identifier that allows linking multiple applications to the same applicant.

In a nutshell, our approach works as follows. Using application-level mortgage data from a confidential version of the widely known HMDA dataset[1], we first split the data into *partitions*, characterized by the distinct outcomes of nine categorical variables, such as census tract of the property or the race of an applicant. We then apply a clustering algorithm to further break down these partitions into *clusters*. These clusters have the property that all applications within one cluster are "close"

---

[1] We discuss our data in more detail in Section 4. For more background, see https://ffiec.cfpb.gov/.

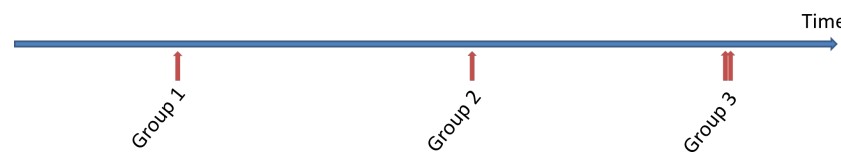

Figure 1: Illustration of clustering algorithm in a hypothetical univariate example for a partition with four applications, where the only clustering variable is "Time." The last two applications are filed within a short time span, and thus assigned to the same cluster.

in terms of a number of continuous attributes, such as application date or reported income. This is motivated by the fact that an applicant may submit two (almost) identical applications to different lenders on subsequent days. We then use our proposed evaluation criteria to optimize our algorithm for precision and sample size. Figure 1 illustrates our approach. Here, we have created a partition of size four using categorical variables in a first step. In the second step, we then use a single continuous variable, for instance application date, to further split the partition into three clusters. The first two clusters consist of single applications (reflecting two applicants), while the third cluster consists of two applications (reflecting a third applicant). Here, the last two applications are assigned to the same cluster because they are filed within a short time span. At this point, all applications in a given cluster are "near-identical." In both simulations and our empirical application, we provide evidence that clusters constructed this way indeed mostly represent single individuals submitting multiple (near-identical) mortgage applications for the same property.

The contributions of this paper are thus twofold. First, we develop a methodological framework to quantify and obtain performance guarantees on arbitrary ML-based anonymous record linkage algorithms in the presence of structural constraints; our results are given in the form of bounds on precision, relative recall, and their weighted linear and harmonic means, and the framework is flexible and can be applied to a variety of datasets where individual-level identifiers are absent. Second, we instantiate our method using a state-of-the-art agglomerative clustering algorithm, making it feasible to apply to large datasets with millions of observations, and apply it to the mortgage market setting, in which consumers can take out only a single first-lien mortgage for a given property. We use our novel evaluation criteria to tune the parameters of our algorithm to optimize its performance, balancing precision with sample size (i.e., relative recall). At our preferred specification of the algorithm, we successfully identify cross-applicants with an estimated 92.3% precision, which underscores the potential of our approach.

## 2 METHODOLOGY

We assume access to a loan-level dataset without personal identifiers. For concreteness, we will frame our following discussion around a dataset of mortgage applications, in line with our empirical application. There are $N$ borrower-property pairs, which we refer to as "individuals." If the same person applies for a mortgage for two distinct properties, we would count this as two distinct individuals. Each individual is indexed by $i$, where $i \in 1, 2, ..., N$. Individuals submit loan applications. They may submit multiple applications for the same property, $m \in 1, 2, ..., n_i$. The number of applications, $n_i$, for each individual may be dependent on the individual's characteristics and can be random.

The covariate vector $Z_{im} = [X_{im}, C_i]$ includes variables observable by the researcher. $X_{im}$ is a vector of variables that may vary across applications if an individual submits multiple applications (e.g., the application date), while $C_i$ is a vector of variables that are constant across an individual's applications (e.g., the census tract for applications involving the same property). The binary indicator $L_{im}$ is set to 1 if individual $i$'s loan application $m$ is accepted, and 0 otherwise. For each approved loan, the applicant decides whether to originate or not. Let $O_{im} = 1$ if a loan is originated, and $O_{im} = 0$ otherwise. We stress that, since we are considering first-lien mortgages throughout, an individual can originate at most one loan.

The researcher does not observe the individual index $i$. Instead, they observe an application $j \in 1, 2, ..., J$, where $J \geq N$ represents a row in the data and consists of $X_j, C_j, L_j$, and $O_j$.

## 2.1 Predicting Labels

While our theoretical bounds introduced below apply to any algorithm that generates predicted labels (personal identifiers), in both our simulations and application we impute the labels using an algorithm based on hierarchical clustering. We thus introduce this algorithm first. Our cross-applicants are constructed as follows. We first split the data into partitions based on the realizations of variables in $C_j$ that are assumed to be constant across applications submitted by the individual $i$ (e.g., census tract). We then further break down these partitions into groups based on the variables in $X_j$ that may differ across applications submitted by the individual $i$ (e.g., date) but expected to be "close." In particular, we group applications such that, for all applications $x_j$ and $x_{j'}$ in the same group, $d(x_j, x_{j'}) \leq \varepsilon$. Here, $d(\cdot)$ denotes distance, $x_j$ is a vector of observed variables for application $j$ of dimension $r$, and $\varepsilon$ is a tuning parameter that determines the maximum distance between two applications in the same group.[2]

**Definition 1.** *We call a cluster of applications $\mathcal{S}$ $\varepsilon$-identical if $d(x_j, x_{j'}) \leq \varepsilon$ and $c_j = c_{j'}$ for any applications $j, j' \in \mathcal{S}$, where $z_j = [x_j, c_j]$ is a vector of observed variables for application $j$.*

We then treat applications in the same cluster as if they were submitted by the same individual, $i$. Our hope is that these clusters indeed represent individual applicants who submitted multiple $\varepsilon$-identical applications.

We first note that finding all $\varepsilon$-identical applications in the data is computationally challenging. A simple strategy is to begin with each application as its own clusters. Then, we iteratively merge the two closest clusters until only one cluster remains. This process results in an inverse tree structure where applications progressively merge into larger clusters. We then select the clusters where all applications within a given cluster are $\varepsilon$-identical by truncating the inverse tree structure at a specific $\varepsilon$ value. All clusters constructed in this manner contain applications $j, j'$ that are identical in terms of their categorical variables ($c_j = c_{j'}$) and near-identical in terms of their continuous variables ($d(x_j, x_{j'}) \leq \varepsilon$). Figure 2 illustrates this process for three applications with identical $c_j$ values (e.g., the same location) but differing loan amounts, denoted as $x_j$. In this example, as we increase $\varepsilon$, the algorithm will first cluster the first two applications together into one $\varepsilon$-identical cluster. Once $\varepsilon$ is larger than 10K, all three applications are clustered into a single cluster.

It is also important to note that once this inverse tree is created, there is no need to recompute clusters for different choices of $\varepsilon$, which facilitates optimally choosing $\varepsilon$ without incurring additional computational costs. The algorithm described above is a version of what is known as agglomerative clustering. Unfortunately, this algorithm has a worst-case time complexity of $\mathcal{O}(\ell^3)$, where $\ell$ is the size of the largest partition. Instead, we apply a state-of-the-art hierarchical agglomerative clustering algorithm for complete-linkage clustering that is based on the nearest-neighbor chain method. This algorithm has a worst-case complexity of $\mathcal{O}(\ell^2)$ while converging to the same clusters that the original, slower algorithm would produce (Müllner, 2011). We implement the agglomerative clustering algorithm using the `fastcluster` package in Python (Müllner, 2013).

## 2.2 Precision and Recall Bounds

In practice, our clustering algorithm may pick up some pairs of applications that are near identical in terms of their observable characteristics, but in fact correspond to multiple individuals. We next propose a way to lower-bound the performance of our algorithm in identifying clusters of applications belonging to the same applicant. In addition, we will use these bounds to choose our tuning parameters ($d(\cdot), \varepsilon$). We make the following assumptions:

**Assumption 1** (Origination decisions are independent across borrowers). *For any $l, m$ and $i \neq j$:*

$$\Pr[O_{im} = 1 | O_{jl} = 1] = \Pr[O_{im} = 1].$$

---

[2]In our empirical implementation, we use a distance function $d(\cdot)$ of the following form:

$$d(x_j, x_{j'}) = \left( \sum_{s=1}^{r} d_s(x_{sj}, x_{sj'})^2 \right)^{1/2}.$$

Note that this corresponds to a weighted $\ell_2$-norm if $d_s(x_{sj}, x_{sj'}) = w_s(x_{sj} - x_{sj'})$, although we also consider more general distances (see Appendix B).

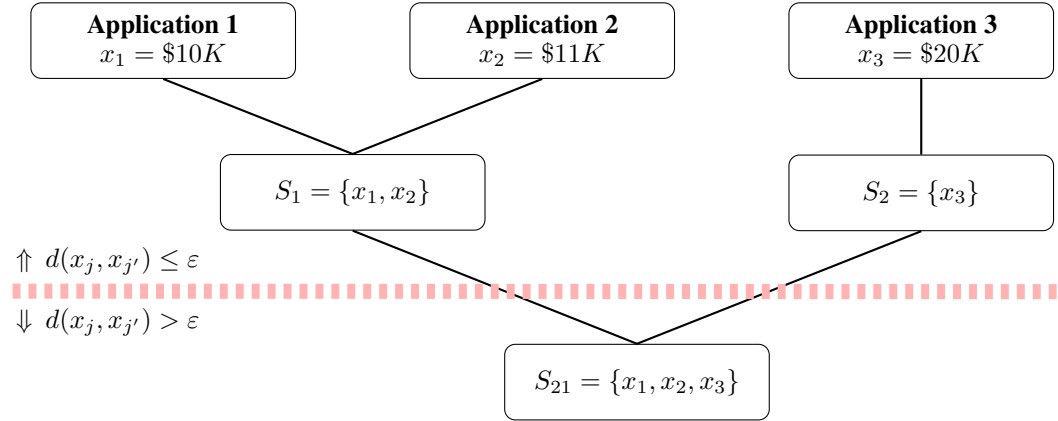

Figure 2: The figure illustrates a hierarchical clustering process for three applications with identical $c_j$ values (e.g., the same location) but differing loan amounts, denoted as $x_j$. The red dashed line represents the cutoff defined by $\varepsilon$ for a hypothetical value of $\varepsilon$ between \$1K and \$10K. The clusters $S_1 = \{x_1, x_2\}$ and $S_2 = \{x_3\}$ are $\varepsilon$-identical clusters at this cutoff. If $\varepsilon$ is increased beyond \$10K (i.e., the red line is lowered), the applications merge into a single $\varepsilon$-identical cluster $S_{21} = \{x_1, x_2, x_3\}$. This example demonstrates how hierarchical clustering relies on forming an inverse tree structure, after which identifying $\varepsilon$-identical clusters becomes straightforward by simply adjusting the position of the cutoff line.

**Assumption 2** (Origination probability is weakly increasing). *An applicant's origination probability is weakly larger in the number of submitted applications:*

$$Pr[\sum_m O_{im} = 1 | n_i = k + 1] \geq Pr[\sum_m O_{im} = 1 | n_i = k] \, \forall k$$

Let `False` denote the event that a cluster is a false positive (i.e. contains applications from more than one applicant), and let `Mult` denote the event that there are multiple originations in a cluster $S$ (i.e. $\sum_{i,m \in S} O_{im} > 1$). Let $p$ be the unconditional probability of origination for an application (i.e. $p = \Pr[O_{im} = 1]$). We obtain the following result.

**Theorem 1.** *Under Assumptions 1-2, the false positive rate can be bounded above as follows:*

$$\Pr[\textit{False}] \leq \frac{\Pr[\textit{Mult}]}{p^2}.$$

*Equivalently, this implies that the precision of our algorithm is at least $1 - \frac{\Pr[\textit{Mult}]}{p^2}$.*

The proof can be found in the Appendix. For intuition, consider a simplified version of the problem where all clusters are at most of size two and applicants submit at most two applications. Recall that no individual can take out two first lien mortgages for the same property: for all $i$, $\sum_{m=1}^{M} L_{im} O_{im} \leq 1$. If our algorithm works perfectly, and each cluster contains only applications from a single applicant, the probability of seeing two originations in the same cluster is equal to zero, since for all clusters $S$: $\sum_{i,m \in S} L_{im} O_{im} \leq 1$. On the other extreme, suppose our clusters contain random pairs of applications. In that case, the probability of seeing two originations in a given cluster can be approximated by $P(O_{im})P(O_{jk}) = P(O_{im})^2$. Thus, the rate at which our algorithm creates clusters with multiple originations is informative about the quality of the algorithm. This allows us to not only assess the quality of our clustering algorithm for a given value of tuning parameters $(d(\cdot), \varepsilon)$ but also choose our tuning parameters.

Theorem 1 means that we can lower-bound the precision of our algorithm using only an estimate of the unconditional probability of origination $p$, and the rate at which our estimated clusters contain multiple originations $\Pr[\texttt{Mult}]$. To estimate these, we simply use the empirical probability of origination in our dataset, $\hat{p}$, to estimate $p$, and the fraction of clusters that have multiple originations, $\hat{p}_m$, to estimate $\Pr[\texttt{Mult}]$.

**Remark 1.** *Note that we can express the Probability of a false positive (cluster with applications from multiple distinct individuals) as*

$$\Pr[\mathtt{False}] = \frac{\Pr[\mathtt{Mult}]}{\Pr[\mathtt{Mult|False}]},$$

*because* $\Pr[\mathtt{Mult}] = \Pr[\mathtt{False}]\Pr[\mathtt{Mult|False}] + \Pr[\neg\mathtt{False}]\Pr[\mathtt{Mult|\neg False}]$, *and* $\Pr[\mathtt{Mult|\neg False}] = 0$ *since we consider first-lien mortgages and therefore an individual can originate at most one loan.*

*Since the empirical counterpart of* $\Pr[\mathtt{Mult}]$ *is observable, the key to Theorem 1 is thus to either know or bound* $Pr[\mathtt{Mult|False}]$.

*Since we restrict ourselves to clusters of size two in both simulations and the application (cf. footnote 4) it may be reasonable to assume that* $\Pr[\mathtt{Mult|False}] = p^2$ *(the probability of multiple originations in a cluster that contains several distinct individuals is equal to the squared unconditional probability of origination for an application) Then, the inequality in Theorem 1 immediately becomes an exact equality.*

*Theorem 1 is more general (i.e., it does not impose* $\Pr[\mathtt{Mult|False}] = p^2$*), but Lemma 1 in the Appendix shows that, under Assumptions 1 and 2,* $\Pr[\mathtt{Mult|False}] > p^2$*. Thus, that Assumptions 1 and 2, which do not appear very strong to us, are still sufficient to provide a lower bound.*

Finally, we note that we can exclude any clusters that indeed contain multiple originations (which we know are false positives) to improve the precision of our algorithm. Our initial clusters contained three types of clusters: 1) True cross-applicants $\hat{\mathcal{S}}^T$, 2) False positives with zero or one originations $\hat{\mathcal{S}}^F_{0-1}$, and 3) False positives with multiple originations $\hat{\mathcal{S}}^F_{2+}$. Since we can easily identify the clusters in the third category, we can improve the performance of our algorithm by simply dropping estimated clusters with multiple originations. This yields a new lower bound on the precision of our algorithm:

$$\Pr[\mathtt{False}] \geq \frac{1 - \frac{\Pr[\mathtt{Mult}]}{p^2}}{1 - \Pr[\mathtt{Mult}]}. \tag{1}$$

Its empirical counterpart is then given by

$$\widehat{\Pr[\mathtt{False}]} \geq \frac{1 - \frac{\hat{p}_m}{\hat{p}^2}}{1 - \hat{p}_m} = \hat{\alpha}(\theta). \tag{2}$$

The following two corollaries to Theorem 1 provide additional lower bounds – one for relative recall and one for weighted summaries of precision and recall. The proofs can be found in the Appendix. Let $\hat{\alpha}(\theta)$ be the lower bound on precision implied by Theorem 1 for tuning parameter $\theta$. Denote by $N^+(\theta) = TP(\theta) + FP(\theta)$ the number of applications the classifier flags as cross-applicants, where $TP(\theta)$ and $FP(\theta)$ are the counts of true and false positives.

**Corollary 1** (Recall bound).

$$Recall(\theta) \geq \hat{\alpha}(\theta)\frac{N^+(\theta)}{P_{tot}},$$

*where $P_{tot}$ is the true number of cross-applicants.*

Since $P_{\text{tot}}$ does not depend on $\theta$, the lower bound on recall is proportional to $\hat{\alpha}(\theta)N^+(\theta)$. Hence, ranking specifications by this bound is equivalent to ranking them by the fully *observable* quantity $\hat{\alpha}(\theta)N^+(\theta)$. We obtain a similar result for the weighted summaries of precision and recall:

**Corollary 2** (Bounds on weighted summaries of precision and recall). *For any $\lambda > 0$, the weighted precision-recall score satisfies*

$$W_\lambda(\theta) = \lambda\, Precision(\theta) + Recall(\theta) \ \geq \ \hat{\alpha}(\theta)\left[\lambda + \frac{N^+(\theta)}{P_{tot}}\right].$$

*Similarly, for any $\beta > 0$, the standard $F_\beta$-score satisfies*

$$F_\beta(\theta) = \frac{(1 + \beta^2)Precision(\theta)Recall(\theta)}{\beta^2 Precision(\theta) + Recall(\theta)} \ \geq \ \frac{(1 + \beta^2)\hat{\alpha}(\theta)N^+(\theta)}{\beta^2 P_{tot} + N^+(\theta)}.$$

Because $P_{\text{tot}}$ is fixed across all $\theta$, these inequalities yield computable lower bounds that can be *maximized* over $\theta$ in both cases using only observable quantities, $\hat{\alpha}(\theta)$ and $N^+(\theta)$.

## 3 SIMULATION

We next generate a hypothetical dataset to illustrate our algorithm and demonstrate its performance in a stylized setting. While we describe our simulated data in more detail in Appendix D, we provide a brief overview here.

We first create one million "census tracts." For each census tract $c$, the number of applicants belonging to this census tract, $N_c$, is random and its distribution approximates the distribution of partitions we observe in our empirical application. The number of applications per applicant $i$ is also random, with expected number of applications per applicant $n_i$ equal to 1.25. In addition to the census tract $C_i$, each application $A_{im} = \{C_i, G_i, T_{im} X_{im}\}$ consists of a discrete group membership $G_i \in \{0, 1\}$ (e.g., gender of the applicant), a continuous variable $T_{im}$ (e.g., time of application), and a continuous variable $X_{im}$ (e.g., the loan amount). $T_{im}$ and $X_{im}$ differ (slightly) across applications $m$ to reflect the observed data. Potential lenders make a decision whether to extend the loan. An applicant hears back sequentially from her applications. As long as she has not originated a loan, each time an application is approved the applicant originates the corresponding loan with probability 0.9. Importantly, once she originates her first loan, the applicant does not originate any additional loans. We reemphasize that the researcher observes application-level data but not the index $i$. That is, she does not know whether two applications $j$ and $j'$ are submitted by the same individual $i$.

### 3.1 RESULTS

We run our partitioning and clustering algorithm on the simulated data, using simple Euclidean distance, $d(x_j, x_k) = \|x_j - x_k\|_2$, as our distance function between two applications $x_j, x_k$ when computing our clusters. To demonstrate how additional observed covariates impact our results, we run our algorithm twice. In the first specification ("without date"), we use only a single continuous variable in $X_j$ during the clustering step, withholding the second observed covariate $T_j$. In the second specification ("with date"), we use both $X_j$ and $T_j$ during the clustering step.[3]

We first illustrate the performance of our algorithm as a function of the tuning parameter $\varepsilon$, where $d(x_j, x_k) < \varepsilon$ for all applications $j$ and $k$ in the same cluster $\mathcal{S}$. Figure 3a depicts the fraction of clusters that consist of applications from a single applicant. This represents the precision (= True Positives/(True Positives + False Positives)) of our algorithm, where a positive instance corresponds to cross-applicants.[4] We first note that the availability of an additional covariate greatly improves the performance of the algorithm: Without the date variable $T_j$, the precision of our algorithm is below 70% for all values of $\varepsilon$, while including the date variable can lead to a precision above 95%. Next, we note that the quality of our algorithm increases as we reduce the size of $\varepsilon$. Intuitively, as we require applications within a cluster to be closer to identical, we reduce the number of "false positives" - applications that look similar but are submitted by distinct applicants. On the other hand, Figure 3b illustrates how the number of estimated cross-applicants increases with $\varepsilon$. We thus face a trade-off - while small values of $\varepsilon$ tend to lead to fewer false positives, we need $\varepsilon$ large enough to obtain a good sample size. In both our specifications, we observe a sweet spot at around $\varepsilon = 0.06$, where our algorithm finds more than 370,000 clusters that contain multiple applications in both specifications, and around 95% ("with date") and 64% ("without date") of these clusters contain only applications from a single applicant.

Since the construction of Figure 3a requires knowledge of an individual's identifier $i$, it is infeasible in practice and cannot be used to assess the quality of the estimates or to choose the value of the tuning parameter $\varepsilon$. However, as we showed in Section 2, the rate at which we find multiple originations in the same cluster is informative about the quality of our algorithm. The probability of origination for a given approved application is $\hat{p} = 0.7917$, and the probability of a random pair of approved

---

[3]Alternatively, we note that the two specifications can be viewed as using two different weight vectors on the covariate vector. That is, with $x_j = [X_j, T_j]$, and using a weighted $\ell_2$-norm of the form $d_w(x_i, x_j) = \left(\sum_{k=1}^{r} w_k (x_{ki} - x_{kj})^2\right)^{1/2}$ as the distance metric between applications, our two specifications ("with date" and "without date") correspond to weight vectors of $w = (1, 1)$ and $w = (1, 0)$, respectively. We will revisit this interpretation in Section 4.

[4]To keep the discussion as simple as possible, we drop all clusters with more than two applications in both our simulation results and our application that follows, such that all results are based on clusters with two applications.

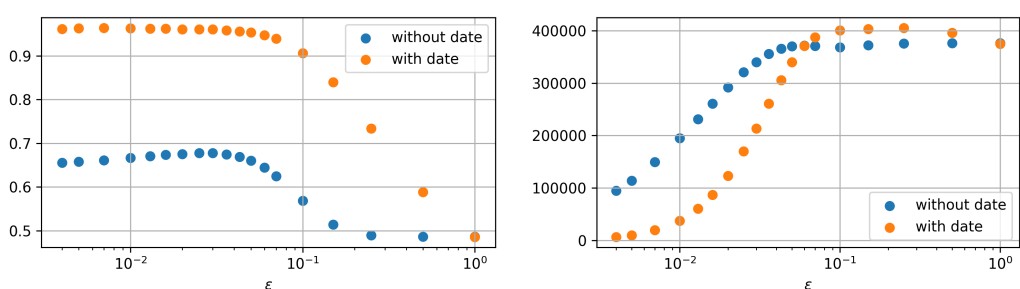

(a) Fraction of clusters that consist of applications from a single applicant (precision).

(b) Number of estimated clusters.

Figure 3: Estimated cross-applicants as function of tuning parameter $\varepsilon$. Raw cluster estimates are adjusted by dropping clusters with multiple originations.

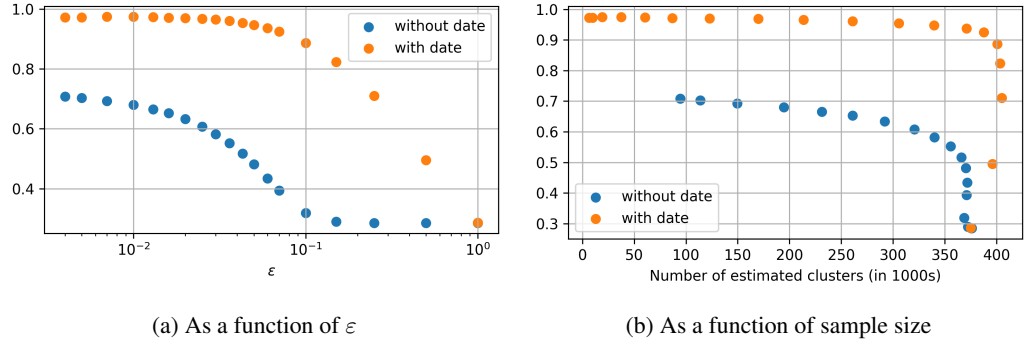

(a) As a function of $\varepsilon$

(b) As a function of sample size

Figure 4: Implied fraction of clusters that consists of true cross-applicants (precision).

applications having both loans originated is thus equal to $\hat{p}^2 = 0.6269$. For a given choice of $\varepsilon$, we can further compute the empirical fraction of clusters with multiple originations, $\hat{p}_m$. Using Equation (2), we can then lower-bound the fraction of clusters that consist of applications from the same applicant. This is depicted in Figure 4a. We first note the close resemblance between Figures 3a and 4a. We take this as an encouraging sign that our proposed method to assess the performance of our algorithm works well. In our main specification ("with date"), a feasible estimate for a lower bound of how many clusters contain applicants from a single applicant is equal to 93.7% at $\varepsilon = 0.06$.

Finally, we directly depict the trade-off between the estimated precision of our algorithm and the number of estimated cross-applicants in Figure 4b. This allows for an easy illustration of the relevant trade-off across the different tuning parameters (e.g., the tolerance $\varepsilon$ and distance function $d(\cdot)$). For example, when considering two sets of tuning parameters $(\varepsilon_1, d_1(\cdot))$ and $(\varepsilon_2, d_2(\cdot))$, we strictly prefer $(\varepsilon_1, d_1(\cdot))$ to $(\varepsilon_2, d_2(\cdot))$ if it results in both higher implied precision and a larger sample size (and thus recall). In particular, going from $\varepsilon = 0.06$ to $0.07$ increases sample size slightly while cutting implied precision by 1 pp. In line with Corollary 1, the same diminishing-returns pattern holds for recall. Hence, the knee at $\varepsilon = 0.06$ maximizes the bound $\hat{W}(\theta)$ for a range of implicit weights $\lambda$ and strikes the best balance. We also note that, at our preferred specification ($\varepsilon = 0.06$) we achieve a recall of 92% (also see Table 2 in the Appendix).

## 4 APPLICATION TO THE US MORTGAGE MARKET

### 4.1 DATA

We obtain data on mortgage applications from the Home Mortgage Disclosure Act (HMDA). The vast majority of all mortgage applications filed in the US are subject to HMDA reporting and thus are included in this dataset. While a publicly available version of this dataset exists, we work directly with

a confidential version (cHMDA) that includes more detailed information for each loan application (e.g., the exact date an application was filed).

We restrict our analysis to mortgage applications filed between 2018 and 2023. We exclude applications from earlier years because a number of important borrower and loan characteristics, such as the credit score and the loan-to-value ratio (LTV) are available only starting in 2018. We further retain only first-lien mortgages and applications that are either approved or denied, dropping applications that are withdrawn by the applicant before a decision was made, applications closed for incompleteness, loan purchases, and applications that went through only the pre-approval process. Finally, we drop applications filed outside the 50 states and Washington, D.C. This gives us 65.5 million applications to analyze. Since the HMDA dataset is at the application level, and not the individual level, it is generally not possible to identify applicants that submit multiple applications. We apply our proposed method to identify these "cross-applicants" who applied for several loans during the mortgage application process.

Following our earlier discussion, we first split the data into partitions based on categorical characteristics we expect to be constant at the individual level. These are: census tract, property type, occupancy, loan purpose, applicant race, applicant sex, applicant age, loan type and a flag for whether or not there is a co-applicant.[5]

Next, we apply the above-mentioned hierarchical agglomerative clustering algorithm to further break down the partitions into clusters. We define our clusters such that, for all applications $x_j, x_{j'}$ in the same cluster the following holds:

$$d(x_j, x_{j'}) \leq \left( \sum_{s=1}^{r} d_s(x_{sk}, x_{sj})^2 \right)^{1/2} \leq \varepsilon, \tag{3}$$

where $x_j$ is a vector of observed variables for application $j$. Specifically, we use $x_j = (date_j, inc_j, size_j, fico_j, ltv_j)$, where $date_j$ represents the date an application is filed, $inc_j$ is the reported income in thousands of dollars, $size_j$ is the requested loan amount in thousands of dollars, $fico_j$ denotes the reported credit score at the time of application, and $ltv_j$ is the loan-to-value ratio of the loan. Note that this corresponds to a weighted $\ell_2$-norm if $d_s(x_{sj}, x_{sj'}) = w_s(x_{sj} - x_{sj'})$, although we also consider more general distances (for more implementation details, see Appendix B).

We consider a total of 96 combinations of distance functions $d(\cdot)$ and tolerance parameters $\varepsilon$, and select the best combination based on an accuracy-sample size trade-off. Figure 5 depicts the precision of our algorithm relative to the sample size for the points $(d(\cdot), \varepsilon)$ on the "frontier": the combinations of $d(\cdot)$ and $\varepsilon$ that are not dominated by an alternative combination yielding both higher precision and a larger sample size. Each point on the curve represents a specific combination of distance function and tolerance parameter. By Corollary 2, maximizing $W(\theta)$ amounts to picking the point on the $(N^+, \hat{\alpha})$ frontier whose slope equals the implicit weight $\lambda P_{\text{tot}}$. Figure 5 (original draft) displays that frontier; our chosen model (orange dot) sits at its knee, where any additional increase in precision would cost a disproportionate loss in sample size (and, by Corollary 1, thus also in recall).

We highlight our preferred specification as the larger orange dot. At this specification, we obtain 314,344 clusters, and estimate that 92.3% are true cross-applicants. We perform additional diagnostics to validate that the clusters truly correspond to cross-applicants in the Appendix.

## 5 CONCLUSION

We presented a novel clustering-based algorithm designed to detect cross-applicants in large anonymized datasets, such as loan-level mortgage data. In particular, our approach introduces a new evaluation method that enables a researcher to optimize the trade-off between precision and sample size (or recall) without the need for labeled training data, making our proposed method highly

---

[5]Note that this implies that applicants who apply for different types of loans cannot end up in the same cluster. This approach makes sense when we want to investigate whether and how a lender distinguishes between two (near-)identical applications submitted by the same individual. In a different context, it may be more appropriate to track an applicant who applies for different types of loans for the same property. Hence, this is an application-specific modeling choice, and we suggest carefully selecting the variables in the distance function to ensure that the identified cross-applicants are consistent with the research question.

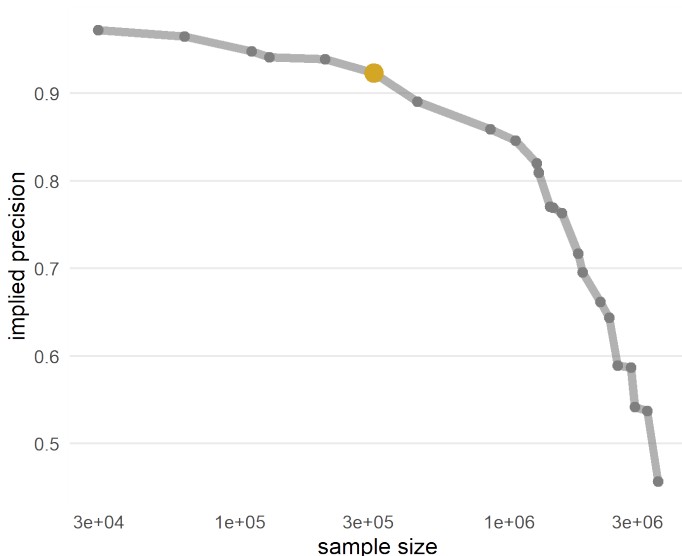

Figure 5: The precision-sample-size frontier for identifying cross-applicants. Each point on the curve represents a specific combination of distance function and tolerance parameter. The large orange dot corresponds to our preferred specification, balancing precision and sample size.

practical. By applying this methodology to the Home Mortgage Disclosure Act (HMDA) dataset, we successfully identified individuals submitting multiple mortgage applications, achieving an estimated precision of 92.3%. Our results open several promising directions for future research. We conclude by highlighting three potential applications of our work in the context of our application:

**1. Measuring Fairness**. Cross-applicants may be useful for measuring fairness across demographic groups in the mortgage market. Elzayn et al. (2025) argue that cross-applicants who had one application approved and a second (near-identical) application rejected can be thought of as "marginal applicants": those on the lenders' decision boundary. Comparing the subsequent default probabilities of these marginal applicants can thus avoid the issue of inframarginality (cf. Simoiu et al. (2017)), and be used to test for discrimination in the lenders' loan granting decisions.

**2. Monitoring Lending Standards and Comparing Banks/Lenders**. Identifying marginal borrowers may also provide a way to better monitor current lending standards. Monitoring the characteristics (e.g., credit scores) of approved applicants over time to measure lending standards (e.g., Mayer et al. (2009), Demyanyk & Van Hemert (2011)) again faces the problem of inframarginality. Monitoring the characteristics of marginal applicants avoids this issue, allowing us to track how lending conditions vary over time or to compare lending standards across lenders.

**3. Exploring Mortgage Shopping Behavior**. Access to a dataset of cross-applicants enables studying the shopping behavior of mortgage applicants (e.g., understanding of how borrowers compare lenders, and how these behaviors differ across various demographic or economic groups). Existing attempts have had to rely on either survey data (Bhutta et al. (2020)) or credit inquiries as proxies for shopping behavior (Agarwal et al. (2024)).

More broadly, our framework is both domain- and method-agnostic. Because our newly derived bounds depend only on observables and non-varying structural terms, they apply to any label-generating algorithm and can be used for hyper-parameter tuning and model comparison without ground-truth labels. Beyond mortgages, the same structural constraints arise in settings such as secured loans, insurance policies, college admissions, or job offers, making the approach widely applicable in cross-institutional or privacy-constrained datasets.

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

## A  Mathematical Appendix

### A.1  Proof of Theorem 1

In this section, we derive how we can use the observed rate of multiple originations per cluster to bound the rate at which applications from distinct individuals are incorrectly clustered together in more detail.

Let $K$ be the number of distinct applicants in a cluster. Before the main result, we start with the following useful lemma.

**Lemma 1.** *The probability of multiple originations in a false cluster is bounded below by $p^2$. That is:*

$$\Pr[\texttt{Mult}|\texttt{False}] \geq \Pr[O_{im} = 1]^2 = p^2.$$

*Proof.* By definition, $K \geq 2$ for false clusters and thus:

$$\Pr[\texttt{Mult}|\texttt{False}] = \Pr[K=2|\texttt{False}]\Pr[\texttt{Mult}|K=2] + \Pr[K>2|\texttt{False}]\Pr[\texttt{Mult}|K>2]$$
$$= \Pr[K=2|\texttt{False}]\Pr[\texttt{Mult}|K=2] + (1 - \Pr[K=2|\texttt{False}])\Pr[\texttt{Mult}|K>2].$$

We can further separate the clusters with two distinct applicants into those that contain exactly two applications, and those that contain more than two applications, and thus write $\Pr[\texttt{Mult}|K=2] = w_2\Pr[\texttt{Mult}|K=2,|\mathcal{S}|=2] + (1-w_2)\Pr[\texttt{Mult}|K=2,|\mathcal{S}|>2]$ where $w_2 = \Pr[|\mathcal{S}|=2|K=2]$. By Assumption 1, $\Pr[\texttt{Mult}|K=2,|\mathcal{S}|=2] = p^2$. Since further, by Assumption 2, $\Pr[\texttt{Mult}|K=2,|\mathcal{S}|>2] \geq \Pr[\texttt{Mult}|K=2,|\mathcal{S}|=2]$, it follows that $\Pr[\texttt{Mult}|K=2] \geq p^2$.

In fact, we can use the same argument to separate the clusters with $K>2$ distinct applicants into those that contain exactly $K$ applications, and those that contain more than $K$ applications. Then, $\Pr[\texttt{Mult}|K=k] = w_k\Pr[\texttt{Mult}|K=k,|\mathcal{S}|=k] + (1-w_k)\Pr[\texttt{Mult}|K=k,|\mathcal{S}|>k]$. By Assumption 1, $\Pr[\texttt{Mult}|K=k,|\mathcal{S}|=k] = 1 - \Pr[\neg\texttt{Mult}|K=k,|\mathcal{S}|=k]$. This probability is given by

$$\Pr[\neg\texttt{Mult}|K=k,|\mathcal{S}|=k] := g(p,k) = (1-p)^k + \binom{k}{1}p(1-p)^{k-1}$$
$$= (1-p)^k + kp(1-p)^{k-1}$$
$$= (1-p)^{k-1}(1-p+kp)$$
$$= (1-p)^{k-1}(1+p(k-1)).$$

With $g(p,k) = (1-p)^{k-1}(1+p(k-1))$, note that for any fixed $p \in [0,1]$,

$$\frac{\partial g}{\partial k} \leq 0$$

for all $k$; to see this, we calculate that:

$$\frac{\partial g}{\partial k} = (1-p)^{k-1}\left[((k-1)p+1)\ln(1-p) + p\right].$$

Then notice that the inner term is 0 at $p=0$ and decreases with $p$, because

$$\frac{\partial}{\partial p}\left[((k-1)p+1)\ln(1-p) + p\right] = (k-1)\ln(1-p) - \frac{pk}{1-p},$$

which is non-positive for any $k \geq 1$, $p \in [0,1]$. Hence $\frac{\partial g}{\partial k}$ is product of a non-negative and non-positive term, i.e. is non-positive overall. In other words, $\Pr[\neg\texttt{Mult}|K=k,|\mathcal{S}|=k]$ is decreasing in $k$. Since $\Pr[\texttt{Mult}|K=k,|\mathcal{S}|=k] = 1 - \Pr[\neg\texttt{Mult}|K=k,|\mathcal{S}|=k]$, we must have that $\Pr[\texttt{Mult}|K=k,|\mathcal{S}|=k]$ is increasing in $k$, and $\Pr[\texttt{Mult}|K=k,|\mathcal{S}|=k] \geq \Pr[\texttt{Mult}|K=2,|\mathcal{S}|=2] = p^2$.

Since, by Assumption 2, $\Pr[\texttt{Mult}|K=k,|\mathcal{S}|>k] \geq \Pr[\texttt{Mult}|K=k,|\mathcal{S}|=k]$, it also follows that $\Pr[\texttt{Mult}|K=k] \geq p^2$. Putting it together, we obtain that

$$\Pr[\texttt{Mult}|\texttt{False}] = \Pr[K=2|\texttt{False}]\Pr[\texttt{Mult}|K=2] + (1-\Pr[K=2|\texttt{False}])\Pr[\texttt{Mult}|K>2]$$
$$\geq \Pr[K=2|\texttt{False}]p^2 + (1-\Pr[K=2|\texttt{False}])p^2 = p^2.$$

$\square$

The proof of Theorem 1 then follows:

*Proof of Theorem 1.* We can write:
$$\Pr[\texttt{Mult}] = \Pr[\texttt{False}]\Pr[\texttt{Mult}|\texttt{False}] + \Pr[\neg\texttt{False}]\Pr[\texttt{Mult}|\neg\texttt{False}].$$
But $\Pr[\texttt{Mult}|\neg\texttt{False}] = 0$ because we consider first-lien mortgages and therefore an individual can originate at most one loan. We can thus write that:
$$\Pr[\texttt{False}] = \frac{\Pr[\texttt{Mult}]}{\Pr[\texttt{Mult}|\texttt{False}]} \leq \frac{\Pr[\texttt{Mult}]}{p^2},$$
where the inequality follows from Lemma 1. It also immediately follows that the precision of our algorithm is given by
$$\text{precision} = \Pr[\neg\texttt{False}] = 1 - \Pr[\texttt{False}] \geq 1 - \frac{\Pr[\texttt{Mult}]}{p^2}.$$
$\square$

## A.2 PROOFS FOR COROLLARY 1 AND 2

*Proof of Corollary 1 (Recall bound).* By definition of precision,
$$\text{Prec}(\theta) = \frac{TP(\theta)}{TP(\theta) + FP(\theta)} = \frac{TP(\theta)}{N^+(\theta)},$$
where $N^+(\theta) = TP(\theta) + FP(\theta)$ is the number of *predicted* positives.

Theorem 1 provides a lower bound
$$\text{Prec}(\theta) \geq \hat{\alpha}(\theta).$$
Multiplying both sides by $N^+(\theta)$ yields
$$TP(\theta) \geq \hat{\alpha}(\theta) N^+(\theta).$$

Recall is the fraction of true positives that are identified:
$$\text{Recall}(\theta) = \frac{TP(\theta)}{P_{\text{tot}}},$$
where $P_{\text{tot}}$ is the (unknown but fixed) total number of true cross-applicants.

Substituting the inequality for $TP(\theta)$ gives
$$\text{Recall}(\theta) = \frac{TP(\theta)}{P_{\text{tot}}} \geq \frac{\hat{\alpha}(\theta) N^+(\theta)}{P_{\text{tot}}} = \hat{\alpha}(\theta)\frac{N^+(\theta)}{P_{\text{tot}}}.$$
This establishes the claimed lower bound. $\square$

*Proof of Corollary 2 (Bounds on weighted summaries of precision and recall).* Recall the definitions
$$\text{Precision}(\theta) = \frac{TP(\theta)}{N^+(\theta)}, \qquad \text{Recall}(\theta) = \frac{TP(\theta)}{P_{\text{tot}}}, \quad N^+(\theta) = TP(\theta) + FP(\theta).$$

The weighted $F$-score is
$$W(\theta) = \lambda\,\text{Precision}(\theta) + \text{Recall}(\theta) = \lambda\frac{TP(\theta)}{N^+(\theta)} + \frac{TP(\theta)}{P_{\text{tot}}}. \tag{1}$$

Theorem 1 implies
$$TP(\theta) \geq \hat{\alpha}(\theta) N^+(\theta). \tag{2}$$
and therefore by inserting (2) into (1), We obtain the following bounds.
$$W(\theta) \geq \lambda\frac{\hat{\alpha}(\theta) N^+(\theta)}{N^+(\theta)} + \frac{\hat{\alpha}(\theta) N^+(\theta)}{P_{\text{tot}}} = \hat{\alpha}(\theta)\Big[\lambda + \frac{N^+(\theta)}{P_{\text{tot}}}\Big].$$

For the $F_\beta$ score, a similar lower bound follows because $F_\beta$ is increasing in $\text{Precision}(\theta)$ and $\text{Recall}(\theta)$. $\square$

# B    IMPLEMENTATION DETAILS

Recall that we use our agglomerative clustering algorithm to break down the partitions of the data into groups such that for all applications $x_j$ and $x_{j'}$ in the same group, $d(x_j, x_{j'}) \leq \varepsilon$. We use a distance function of the following form:

$$d(x_j, x_{j'}) = \left( \sum_{s=1}^{r} d_s(x_{sj}, x_{sj'})^2 \right)^{1/2}.$$

Table 1 summarizes the different ways we compute the distance between applications. Each row corresponds to a specific definition of the distances $d_s(\cdot)$ for $s = 1, \ldots r$. If $d_s$ is numeric, $d_s(x_{sj}, x_{sj'}) = w_s(x_{sj} - x_{sj'})$ with the number in Table 1 indicating the value of $w_s$. Thus, if the entire weight vector is numeric, the corresponding row represents a weighted $\ell_2$-norm. If $w_s$ is equal to "Penalize exact",

$$d_s(x_{sj}, x_{sj'}) = \begin{cases} (x_{sj} - x_{sj'}) & \text{if } x_{sj} \neq x_{sj'} \\ 55 & \text{if } x_{sj} = x_{sj'}. \end{cases}$$

If $w_s$ is equal to "Reward exact",

$$d_s(x_{sj}, x_{sj'}) = \begin{cases} 0 & \text{if } |x_{sj} - x_{sj'}| < 7 \\ 2(|x_{sj} - x_{sj'}| - 7) & \text{otherwise.} \end{cases}$$

Table 1: Universe of distance functions considered. Each line corresponds to one definition of distance between applications.

| | Application Date | Income | Loan Amount | Credit Score | Property Value |
|---|---|---|---|---|---|
| 1. | 1 | 0 | 0 | 0 | 1 |
| 2. | 1 | 0 | 1 | 0 | 0 |
| 3. | 1 | 1 | 0 | 0 | 0 |
| 4. | 1 | 1 | 1 | 0 | 1 |
| 5. | 1 | 1 | 1 | 1 | 1 |
| 6. | 1 | 1 | 1 | 2 | 1 |
| 7. | 1 | 1 | 1 | 3 | 2 |
| 8. | Penalize Exact | Penalize Exact | 1 | 1 | 1 |
| 9. | Penalize Exact | Penalize Exact | Penalize Exact | 1 | 1 |
| 10. | Reward Exact | 1 | 1 | 1 | 1 |
| 11. | Reward Exact | 1 | 1 | 2 | 1 |
| 12. | Reward Exact | 1 | 1 | 3 | 2 |

For each row, we then use $\varepsilon \in \{15, 22, 30, 40, 52, 70, 90, 110\}$ for a total of 96 combinations of $(d(\cdot), \varepsilon)$. Figure 6 shows the precision and sample size for each of these combinations. We run our algorithm on an internal cluster with 64 3500MHz processors with 16 cores each and 377GB memory. The entire algorithm (partitioning the 65.5 million applications in the data, clustering the resulting partitions according to the twelve distance definitions in Table 1, then traversing the resulting trees and calculating the performance for the eight versions of $\varepsilon$) takes less than a day.

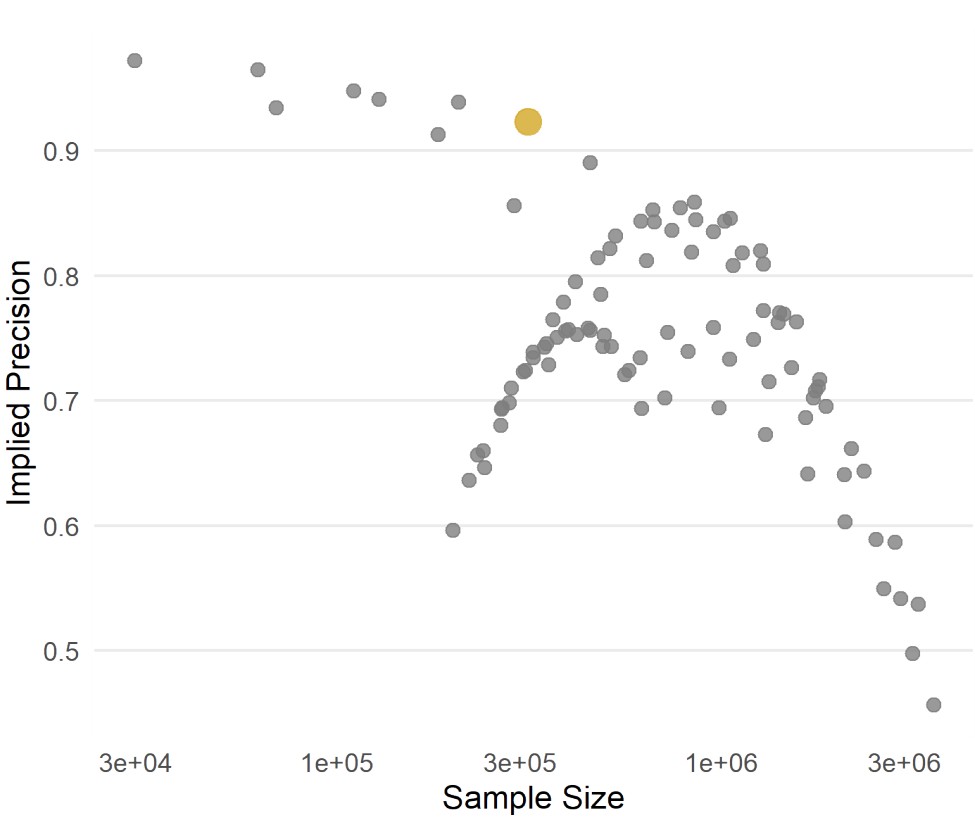

Figure 6: Precision and sample size of our algorithm as a function of different distance definitions and tolerances. Each point on the curve represents a specific combination of distance function and tolerance parameter. The large orange dot corresponds to our preferred specification, balancing precision and sample size.

## C    ADDITIONAL EMPIRICAL RESULTS

Figure 7 provides several diagnostics to validate our estimated cross-applicants. Figure 7a depicts the distribution of the difference in dates between applications within a cluster. We depict this distribution separately for those cross-applicants that have their first application denied and approved. We see that most applications within the same cluster are submitted within 3 weeks, and observe a bunching of the differences at multiples of seven, corresponding to applications being submitted on the same weekday. Further, cross-applicants who first get denied take slightly longer to submit their second application, compared to cross-applicants who have their first application approved. This may suggest that applicants who had their first application denied may try to improve their profile before applying again. In Figures 7b-7d, we therefore explore how some of the key variables that determine a loan decision vary between the two applications an applicant submits.

Figure 7b shows that applicants whose first application is rejected tend to have a slightly higher credit score on their second application, compared with applicants whose first application is approved, when they submit their second application within four weeks of their first application. After four weeks, both groups experience a drop in their credit score. This is consistent with a hard credit inquiry as a result of their first application negatively impacting their credit score.

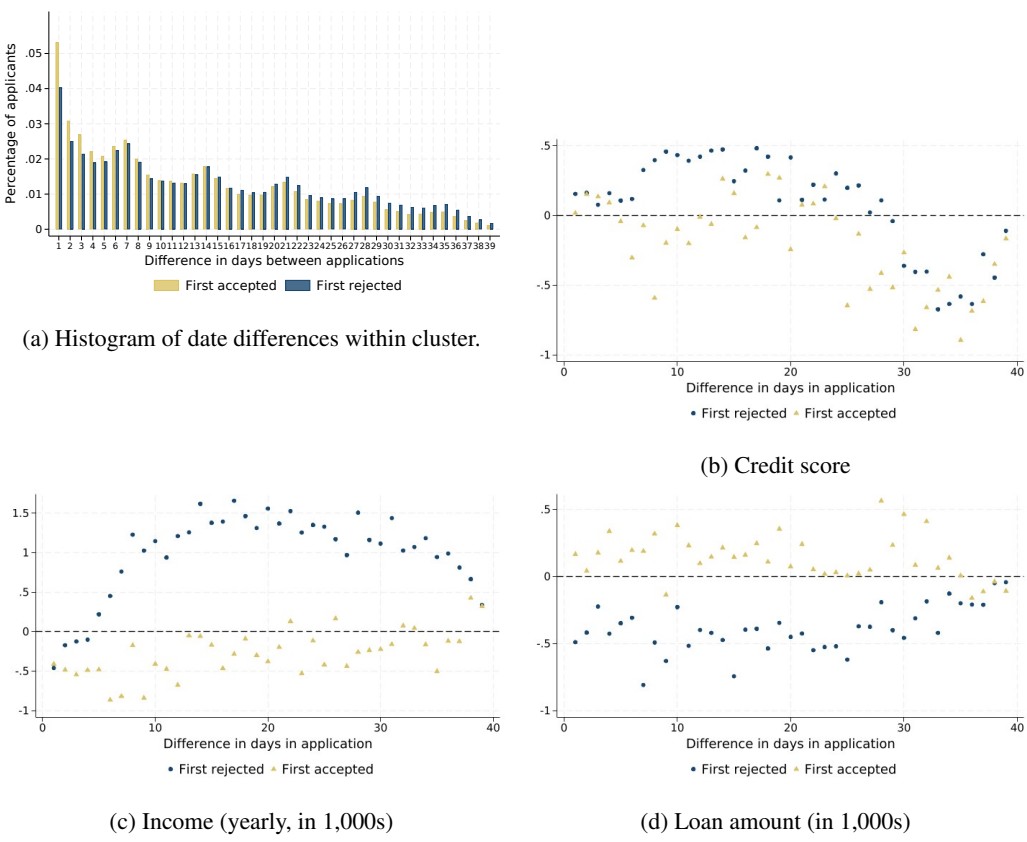

(a) Histogram of date differences within cluster.

(b) Credit score

(c) Income (yearly, in 1,000s)

(d) Loan amount (in 1,000s)

Figure 7: Average change in the reported variable in an applicant's second application relative to her first application, as a function of the date differences between the first and second application submitted. Yellow triangles represent cross-applicants whose first application was approved. Blue dots represent cross-applicants whose first application was denied.

Figure 7c shows that applicants whose first application is rejected tend to report a higher income on their second application, compared with applicants whose first application is approved. This is consistent with denied applicants taking steps to (marginally) improve their profile if their first application is rejected. In particular, we note that income is generally self-reported (though subject to verification), making it potentially easier to alter compared to, for instance, one's credit score.

Figure 7d shows that applicants whose first application is rejected tend to request a slightly lower loan amount on their second application, compared with applicants whose first application is approved. This is again consistent with denied applicants taking steps to (marginally) improve their profile if their first application is rejected.

# D    SIMULATION DETAILS

We first create one million "census tracts." For each census tract $c$, the number of applicants belonging to this census tract $N_c$ is equal to $1 + \psi_c$, where $\psi_c$ drawn from a Poisson distribution with parameter $\lambda = 1$ to approximate the distribution of partitions we observe in our empirical application (cf. Figure 8 below).

Next, an applicant $i$ submits a loan application. After each application, she continues to submit another application with probability $0.2$ such that the expected number of applications per applicant $n_i$ is $1.25$.

We then create features associated with each application (in addition to the census tract $C_i$) as follows. First, to create a second variable that is constant across an individual's applications we randomly assign a group membership $G_i \in \{0, 1\}$ to applicant $i$ with $Pr(G_i = 1) = \gamma_0 \in (0, 1)$, where $G_i$ is independent across $i$. The variable $G_i$ may represent characteristics such as race or gender. Next, each application $m$ by applicant $i$ is associated with three further covariates $X_{im}, T_{im}$ and $\eta_i$. We assume that both $X_{im}$ and $T_{im}$ are observed by the researcher, while $\eta_i$ is not. We stress that $X_{im}$ and $T_{im}$ may differ (slightly) across applications $m$ to reflect the observed data. We create realizations of these random variables as follows. $T_{im}$ may reflect the time of the application, and is equal to $T_{im} = \tilde{T}_i + \nu_{im}$, where $\tilde{T}_i \sim Unif[0, 1]$ and $\nu_{im} \sim N(0, \sigma_T)$. $X_{im}$ may reflect the loan amount, and is equal to $X_{im} = \tilde{X}_i + \xi_{im}$, where $\xi_{im} \sim N(0, \sigma_X)$. The conditional distribution of $(\tilde{X}_i, \eta_i)$ conditional on $G_i$ is given by

$$[\tilde{X}_i, \eta_i]' \sim Lognormal(\mu_g, \Sigma_g), \tag{4}$$

where $\mu_g$ and $\Sigma_g$ are mean and covariance matrix of bivariate normal for $G_i = g$. Specifically, we use $\gamma_0 = 0.5$, $\mu_0 = [-3, -3]$, $\mu_1 = [-2.5, -2.5]$, $\Sigma_0 = \Sigma_1 = [0.25\ 0.1, 0.1\ 0.25]$.

Finally, each applicant has a default behavior associated with her. We assume that the default probability of an applicant, $Pr(D_i = 1)$, depends on both $\tilde{X}_i$ and $\eta_i$ as follows:

$$Pr(D_i = 1 | \tilde{X}_i, \eta_i, G_i, \tilde{T}_i, C_i) = \min(1, \delta_0 + \delta_1 \tilde{X}_i + \delta_2 \eta_i). \tag{5}$$

Potential lenders observe (or are able to estimate) an individual's default probability $P(D_i)$, and their decision whether to extend the loan takes the form:

$$P(L_{im} = 1) = \begin{cases} 1 & \text{if } P(D_i = 1) < 0.27 \\ 0.5 & \text{if } P(D_i = 1) \in [0.27, 0.29] \\ 0 & \text{if } P(D_i = 1) > 0.29. \end{cases} \tag{6}$$

An applicant hears back sequentially from her applications. As long as she has not originated a loan, each time an application is approved the applicant originates the corresponding loan with probability $0.9$. Once she originates her first loan, the applicant does not originate any additional loans.

We reemphasize that the researcher observes application-level data without knowing the index $i$.[6] That is, she does not know whether two applications $j$ and $j'$ are submitted by the same individual $i$. For each application $j$, the variables $G_j, C_j, X_j, T_j, L_j$ as well as the element-wise products $D_j L_j$ and $O_j L_j$ are observable by the researcher. On the other hand, $\tilde{X}_i, \tilde{T}_i$, and $\eta_i$ are unobserved to the researchers.

Finally, we compare the number of observed applications per partition in our simulated data and our empirical application in Figure 8. Both histograms indicate that the majority of partitions is relatively small. This is intuitive: In our application, each partition includes only applications for mortgages in a small geographic region (census tract), and among those we further separate applications by eight additional discrete attributes of applicant and property. For Figure 8b, partitions larger than twelve applications are omitted for clarity. The largest partition in our application contains 153 applications.

---

[6]However, while this is infeasible in practice, knowing the index $i$ in our simulated dataset will allow us to evaluate the performance of our algorithm.

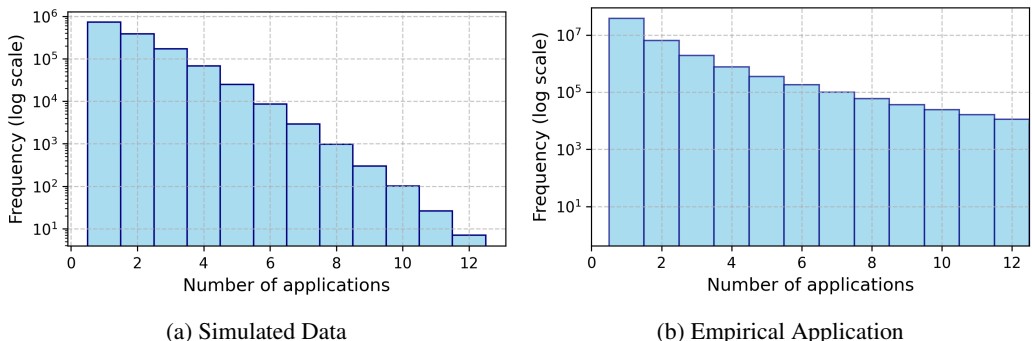

(a) Simulated Data    (b) Empirical Application

Figure 8: Distribution of partition sizes (number of applications per partition). In our empirical application we truncate the histogram at twelve for better readability. Less than 0.1% of partitions include 13 or more applications.

## D.1    ADDITIONAL SIMULATION RESULTS

| $\varepsilon$ | Precision | Recall | Implied Precision ($\hat{\alpha}$) | Sample Size ($N^+$) |
|---|---|---|---|---|
| 0.04 | 0.96 | 0.78 | 0.95 | 266799 |
| 0.05 | 0.96 | 0.85 | 0.95 | 289280 |
| 0.06 | 0.95 | 0.92 | 0.94 | 306898 |
| 0.07 | 0.94 | 0.96 | 0.93 | 314405 |
| 0.10 | 0.92 | 1.00 | 0.89 | 317280 |

Table 2: Summary of performance metrics for different $\varepsilon$ values in our simulation ("with date").

