# OpenReview forum: "Precision Without Labels: Detecting Cross-Applicants in Mortgage Data Using Unsupervised Learning"
_ICLR.cc/2026/Conference — Submitted to ICLR 2026_

### Official Review · Reviewer_8odU · 2025-10-28

**Soundness:** 2
**Presentation:** 2
**Contribution:** 2
**Rating:** 2
**Confidence:** 3

**Summary:**

This paper introduces a novel unsupervised evaluation method for anonymous record linkage, addressing the key challenge of assessing model performance without labeled training data. The approach derives observable lower bounds on precision and relative recall by leveraging a structural constraint namely, the limited number of positive outcomes (matches) a single individual can have. This framework enables principled model selection and hyperparameter tuning for linkage tasks where labels are unavailable. The authors demonstrate their method on the Home Mortgage Disclosure Act (HMDA) dataset, detecting “cross-applicants” who submit multiple loan applications without the use of personal identifiers. Their approach achieves 92.3% precision with minimal relative recall loss, showing strong practical utility for privacy-preserving linkage tasks.

**Strengths:**

1. The work directly tackles the fundamental challenge of evaluating record linkage in anonymized or unlabeled settings, a recurring issue in privacy-preserving data integration.

2. The derivation of lower bounds on observable precision and recall is elegant and grounded in sound probabilistic reasoning, ensuring a strong theoretical foundation.

3. By avoiding the need for labeled data, the approach is highly applicable to sensitive domains (finance, healthcare, census) where privacy laws restrict data linkage supervision.

4. The empirical case study provides a realistic and policy-relevant use case (loan applications), illustrating the method’s feasibility in detecting duplicate or related records.

**Weaknesses:**

1. The evaluation is based solely on HMDA data, which may not represent broader linkage challenges across diverse domains (e.g., healthcare, e-commerce, or census data).

2. The method relies on the assumption that each individual can have only a limited number of positive outcomes, which might not hold in certain contexts (e.g., multiple affiliations or family-linked entities).

3. The paper does not explore how the model behaves under record duplication errors, partial overlaps, or noisy features, which are common in real-world datasets.

4. While “relative recall” is useful when ground truth is unavailable, its interpretability for practitioners is limited compared to traditional recall metrics.

**Questions:**

1. How tight are the proposed lower bounds on precision and recall under different data distributions?
2. Have you analyzed how these bounds behave under class imbalance or varying linkage density?
3. How sensitive is the framework to violations of the “limited positive outcomes per individual” assumption?
4. Could false assumptions lead to overly optimistic or pessimistic estimates of performance?
5. How easily can this method generalize to other types of anonymous linkage problems, such as medical patient records or transactional data, where the constraint structure differs?

---

### Official Review · Reviewer_KBW4 · 2025-10-30

**Soundness:** 2
**Presentation:** 3
**Contribution:** 2
**Rating:** 4
**Confidence:** 3

**Summary:**

This paper proposes a novel framework for evaluating unsupervised record linkage algorithms without labeled training data. The authors derive observable lower bounds on precision and relative recall by exploiting a structural constraint: individuals can have at most one positive outcome (e.g., originate only one first-lien mortgage). They demonstrate their approach on HMDA mortgage data using hierarchical agglomerative clustering to identify cross-applicants (individuals submitting multiple applications), claiming 92.3% precision at their preferred specification.

**Strengths:**

**S1**- Novel Theoretical Contribution: The derivation of observable lower bounds on precision and relative recall without labels (Theorem 1 and Corollaries 1-2) is genuinely innovative. The key insight—exploiting structural constraints where individuals can have atmost one positive outcome, is elegant and addresses a real gap in unsupervised learning evaluation.

**S2**- Domain-Agnostic Framework: The method is algorithm-agnostic and generalizable beyond mortgages to insurance, college admissions, job applications, etc., which significantly broadens its impact.

**S3**- Computational Efficiency: Using the nearest-neighbor chain method for hierarchical clustering with $O(l^2)$ complexity is practical for large datasets.

**S4**- Strong Empirical Validation: The simulation results (Figure 3a, 4a) show close correspondence between true precision and bounded precision, validating the theoretical framework.

**Weaknesses:**

**Criticals**

**CW1**. Assumption 1 (independence of origination decisions across borrowers) ignores supply constraints, market-level shocks, and spatial correlation in mortgage markets. Assumption 2 (monotonically increasing origination probability) contradicts observed credit score degradation from multiple inquiries. **Neither assumption receives empirical validation**.

**CW2**- Theorem 1 provides Pr[False] ≤ Pr[Mult]/p², but the gap between the lower bound (92.3%) and true precision remains unquantifed. Lemma 1 only shows Pr[Mult|False] ≥ p², not equality. the expectation is the simulation data with ground truth should quantfy this gap.

**CW3**- All clusters with >2 applications are dropped without justification. This limitation leaves critical questions unanswered: What fraction of true cross-applicants submit 3+ applications? How does this affect recall? The theory should extend to larger clusters or justify the restriction.

**CW4**- The paper lacks any comparison to established methods: Fellegi-Sunter probabilistic linkage, deep learning entity resolution, alternative clustering algorithms (DBSCAN, spectral clustering), or existing evaluation methods for record linkage without labels. **Cannot assess whether the approach is actually useful**.

**CW5**- Testing 96 combinations of distance functions and tolerance parameters with no principled selection criteria. Partition variables (census tract, property type, race, sex, age) lack sensitivity analysis for misreporting or changes. **No guidance for practitioners on feature selection for new applications**.

**CW6**- Selection Bias from Filtering False Positives: dropping clusters with multiple originations improves precision but creates selection bias—the bound in Equation (1) applies only to the filtered sample, conflating "identifying cross-applicants" with "removing obvious errors." This undermines the evaluation framework's integrity.

**Minors**

**MW1**- Validation Using Behavioral Patterns is Circular: Figure 7 "validates" the results by showing: applications in clusters are close in time (Figure 7a) and denied applicants improve their profiles (Figures 7b-d). But these patterns are exactly what the clustering algorithm was designed to find! This is not independent validation.

**MW2**- No Ground Truth Validation Subsample: The paper has access to confidential HMDA data. Why not:

   - Create a subsample with validated matches (manual review, SSN matching if available)

   - Test the bounds against ground truth

   - Validate Assumptions 1-2 empirically

**Questions:**

Weak points and following questions:

**Q1**- Lemma 1 Proof: The proof shows ∂g/∂k ≤ 0, but the calculation seems to have a sign error. With g(p,k) = (1-p)^(k-1)(1 + p(k-1)), shouldn't the derivative with respect to k include the product rule more carefully?

**Q2**- Why Complete-Linkage?: The paper uses complete-linkage clustering (maximum distance within cluster ≤ ε) but doesn't justify this choice over single-linkage or average-linkage. Complete-linkage is sensitive to outliers—is this desirable?

**Q3**- How are Ties Handled?: When d(xj, xj') = ε exactly, are applications clustered together or separated? This could materially affect results at the boundary.

**Q4**- Corollary 2 Interpretation: The weighted F-score bound depends on unknown Ptot. The paper claims you can "maximize" the bound, but without knowing Ptot, how is this operationalized?

---

### Official Review · Reviewer_HCMw · 2025-10-31

**Soundness:** 2
**Presentation:** 3
**Contribution:** 2
**Rating:** 4
**Confidence:** 3

**Summary:**

The paper proposes a method to evaluate anonymous record linkage without labeled data by exploiting structural exclusivity constraints, such as the one-loan-per-borrower rule for first-lien mortgages. Predicted links are produced by agglomerative clustering within partitions defined by variables presumed constant for an individual, and performance is lower-bounded using the observed frequency of clusters with multiple originations and the unconditional origination rate. The empirical application to confidential HMDA demonstrates that the approach can tune clustering hyperparameters to trade precision against sample size and reports high implied precision at the preferred setting. The simulations + diagnostics help illustrate the mechanism and the practical choices in distance metrics and hyperparameter thresholds.

**Strengths:**

The central idea of converting an exclusivity constraint into observable bounds on precision and relative recall is useful when ground truth is unavailable. The theoretical connection between multiple originations in a cluster and a lower bound on precision is simple but general. The framework is method-agnostic with respect to the label generator, which makes it potentially compatible with many record-linkage pipelines. The empirical exercise is large-scale and carefully engineered, and the precision–sample-size frontier usefully visualizes the tuning trade-off. The application to cross-applicants has clear downstream consequences for auditing and market monitoring.

**Weaknesses:**

- The approach does not, I think, engage much with existing approaches seeking to do something similar (e.g., Enamorado et al. 2019 or the long-standing Bayesian record linkage literature). Some of the methods among those existing approaches could be important points of empirical comparison. It is not clearly how proposed method compares against existing baselines.

- The approach relies on an accurate unconditional origination probability despite likely heterogeneity across observables and time. Because the bound uses a single p, subpopulations with lower approval rates will have fewer multiple-origination events even when merges are spurious, which could make the bound optimistic in those strata.

- The assumptions of independence across borrowers and monotonicity in the number of applications are possibly strong in mortgage settings where tract-, lender-, and time-level shocks could induce correlation; the direction of bias will vary with the sign of these correlations, so sensitivity analyses would be helpful.

- The empirical validation is confined to mortgages. The framing and notation are tailored to this case, which risks overstating generality; the paper would be stronger with a second, unrelated domain where an exclusivity rule holds.

- As noted earlier, the clustering component is narrow relative to the entity-resolution literature. Agglomerative complete-linkage with an L2-style distance is a reasonable baseline, but there is little comparison against alternative linkage strategies such as Bayesian or probabilistic record linkage, graph-based community discovery, or modern embedding-based similarity, which makes it hard to assess performance.

- The practice of dropping clusters with multiple originations both changes downstream estimands and the lower bound on algorithm precision, while introducing researcher discretion; the paper derives an adjusted bound, but it would help to articulate ex-ante decision rules or heuristics.

- The claim that the rate of clustering merges is itself informative about quality is asserted in prose but not fully substantiated empirically; more direct evidence or illustration would clarify how this guidance should be operationalized.

- Finally, the empirical section could be clearer about the nature of second applications—whether they are rejected-then-resubmitted cases or different-lender near-duplicates—since this affects the interpretation of some of the results.

**Questions:**

- How sensitive are the precision and recall bounds to replacing the single (I believe) unconditional origination rate with subgroup-specific or time-varying rates, for example, by tract, month, lender, or applicant score bins, and does this materially change the chosen tuning parameter, epsilon?

- Can the independence assumption be relaxed by allowing within-cell correlation and reporting bounds as a function of an intraclass correlation parameter, along the lines of a worst-case analysis?

- What happens when there are essentially no true matches in a partition or dataset? A negative-control-style exercise would help ensure the procedure does not fabricate cross-applicants.

Can the authors provide head-to-head comparisons with representative alternatives from probabilistic record linkage, Bayesian entity resolution, and embedding-based similarity, or other clustering strategies?

**Details Of Ethics Concerns:**

Linkage without identifiers risks de-identification of anonymous datasets.

---

### Official Review · Reviewer_Ea8u · 2025-11-08

**Soundness:** 2
**Presentation:** 2
**Contribution:** 1
**Rating:** 0
**Confidence:** 5

**Summary:**

The paper proposes a basic distance base method to cluster records and identify mortgages that may be submitted by the same individuals

**Strengths:**

The problem is interesting and some theoretical results are provided.

**Weaknesses:**

The approach is evaluated on only a single dataset, which limits the strength and generalizability of the conclusions. There exists a rich body of literature on the record linkage problem, along with many publicly available benchmark datasets that could have been leveraged to provide a more comprehensive evaluation.

More importantly, there is no comparison with other established baselines. It would be useful to include comparisons with both simple heuristic methods and more advanced approaches, such as those incorporating embedding-based similarity measures, to better assess the effectiveness of the proposed method.

**Questions:**

None.

---

### Meta-Review · Area_Chair_8MQ4 · 2025-12-11

**Summary:**

While the theoretical problem formulation is interesting and potentially general, the current evidence does not meet ICLR standards for novelty and significance demonstrated via rigorous comparison and validation. I therefore recommend reject.

**Reviewer Concerns:**

**Ea8u** single-dataset evaluation, no baseline comparisons.

**HCMw** limited engagement with prior work, reliance on single approval rate, strong independence/monotonicity assumptions, dropping multi-origination clusters, missing alternative linkage methods.

**KBW4** unvalidated assumptions, unquantified gap between bound and true precision, unprincipled hyperparameter sweeps, selection bias from filtering.

**8odU** limited generalization beyond HMDA, robustness to noise/duplication, weak interpretability of relative recall.

No rebuttal submitted by the authors, and all concerns are still outstanding.

**Reviewer Scores:**

No reason at all to increase their scores.

---

### Decision · Program_Chairs · 2026-01-26

Reject